# Left-Behind Children, Parent-Child Communication and Psychological Resilience: A Structural Equation Modeling Analysis

**DOI:** 10.3390/ijerph18105123

**Published:** 2021-05-12

**Authors:** Chi Zhou, Qiaohong Lv, Nancy Yang, Feng Wang

**Affiliations:** 1Medical School, Hangzhou Normal University, Hangzhou 311121, China; zhouchi@hznu.edu.cn; 2Department of Health Education, Zhejiang Provincial Center for Disease Control and Prevention, Hangzhou 310051, China; qhlv@cdc.zj.cn; 3Medical School, University of Minnesota, Minneapolis, MN 55455, USA; yang4948@umn.edu; 4School of Medicine, Zhejiang University, 866 Yuhangtang Road, Hangzhou 310058, China

**Keywords:** left-behind children, parental migration status, parent–child communication, psychological resilience

## Abstract

This study aims to examine the role of parental migration status and parent communication in the psychological and related behavioral status of left-behind children and their psychological resilience. A cross-sectional survey was conducted in Anhui Province of China, and a questionnaire survey was conducted with 1992 teens using the Chinese version of the Parent-Adolescent Communication Scale (PACS), Connor-Davidson Resilience Scale (CD-RISC), and Strength and Difficulties Questionnaires (SDQ). Compared with the never left-behind group, left behind children had relatively lower PACS, CD-RISC and SDQ scores. Absence of parents is related with poorer psychological resilience, while good parent communication is related with better psychological resilience. Better psychological resilience is related to fewer psychological problems regardless of parental migration status. Currently left-behind status demonstrated a negative influence on psychological resilience, while never left-behind status had a positive effect. Interventions are needed to enhance psychological resilience of left-behind children to prevent psychological and related behavioral problems.

## 1. Introduction

Rapid urbanization in China has led to a large number of rural-to-urban migrant workers [1,2]. Many rural migrants have been seeking employment opportunities in urban areas, and their children usually stay in their areas of origin and are often raised by grandparents or relatives. Left-behind children are defined as children who remain in rural regions of China while one or both of their parents leave to work in urban areas for over 6 months, and they were left behind with their grandparents or relatives etc. [3]. A study of Chinese left-behind children showed that there are more than 9,020,000 in rural areas in 2018 year [4]. Previous studies have shown that parental migrant status exerts adverse impacts on the healthy development of children [5]. Left-behind children were more likely to suffer depression and loneliness, had a high risk of suicide, and exhibited poor nutrition and other health problems [1,4,5,6].

### 1.1. Parental Migration Status and Mental Health

The focus of this study is to assess the impact of parental migration on the mental health of left-behind children. Most studies suggest that left-behind children are more likely to have mental health disorders, especially conduct problems such as hyperactivity/inattention, and peer relationship problems [7,8,9]. A cross-sectional survey showed that, in China, the depressive symptom score of rural left-behind children was significantly higher (40.4%) than that of the non left-behind group (27.8%), and the prevalence of personality symptoms was significantly higher (64.4% vs. 47.6%) [10]. However, a minority of studies showed that left-behind children do not have worse mental well-being than other children [11]. Previous studies have also reported risk factors associated with the mental health of left behind children, such as lack of family social capital affecting child interaction and emotional support, the poor utilization of social support [12], lower levels of life satisfaction and parenting efficacy [11]. However, psychological resilience may be a mediating factor on mental health from the perspective of positive psychology [13]. Resilience is defined as resistance, recovery, or rebound of mental health after a challenge, and psychological resilience indicates that resilient people are able to “bounce back” from stressful experiences quickly and efficiently [14,15]. Few studies have assessed the role of psychological resilience as a mediating factor between parental migration status and the mental health of left behind children. One study which explored the psychological resilience of left-behind children in Sichuan province of China found that left-behind children demonstrated significantly lower resiliency scores, leading to psychological problems [16,17].

### 1.2. Parental Migration Status and Parent–Child Communication

Due to long term parental–child separation in rural-to-urban migrant families, left-behind children often have deficient communication with their parents regarding their emotional state, and are prone to develop psychological problems [18,19,20]. A study in Chongqing province in China showed that low frequency of parent–child communication was a risk factor for depression symptoms among left-behind children between the ages of 7–17 years [21]. Some research has indicated that parental support and parent–child communication can enhance the quality of life of left-behind children in China [22,23,24]. One study of 1165 rural left-behind children showed that good parent–child communication was related to greater life and school satisfaction and happiness [25]. Alternatively, with the development of modern communication technology in China, parents can now contact with their children anytime via mobile phone and/or instant-messaging software such as WeChat video (Chinese version of Facebook). This kind of communication depends more on parental will, and is not limited by the poor communication conditions of the past, such as waiting for a reply letter for a long time [26].

### 1.3. The Current Study

In conclusion, there is a great deal of literature on the mental health of left-behind children; however very few studies focus on the effect of psychological resilience, which is a key mediating factor in mental health. Additionally, very few studies have explored the role of parent–child communication in psychological resilience among different parental migration statuses, especially against the current and better communication background (phone calls, video chats vs. letters, etc.). The present study aims to fill this gap by examining how parental migration status and parent-child communication play a role in the mental health of left-behind children and their psychological resilience. This study stands out from prior studies by using structural equation modeling (SEM) in the analysis of three parental migration status models, thus enabling the comparison of associations between the different outcomes and the measurement of indirect effects.

## 2. Materials and Methods

### 2.1. Participants and Procedure

A cross-sectional survey was conducted in April 2018. The participants were recruited from Wuwei and Nanling in Anhui province, which are relatively underdeveloped counties of southeastern China and have large numbers of left-behind children. We randomly selected two towns from each county, and then randomly selected one primary school and one middle school from each town. In each school, students were recruited if they met the following criteria: (1) enrolled in either grade 5–6 in primary school or grade 7–8 in junior middle school, (2) were 11–17 years old, (3) had a local household registration in either Wuwei or Nanling county, (4) were willing to participate in the study, (5) were not suffering from any psychological disorder, and (6) both parents were not deceased, divorced or remarried. In total, 1992 teens from eight schools in four towns were included in this study. All participants signed the consent form and were aware that their participation was purely voluntary. Students were then asked to complete self-administered questionnaires in their classrooms. In order to make sure students could fill in sensitive questions, teachers or school administrators were required not to be present at classroom. Ethical approval was obtained from Zhejiang University and local approvals were obtained from county authorities.

### 2.2. Measurement of Parent-Child Communication

The communication between parent and child was assessed with the Chinese version of the Parent-Adolescent Communication Scale (PACS) [27,28,29]. The scale has 20 items, and includes the open family communication sub-scale (10 items) and the problems in family communication sub-scale (10 items). The open family communication sub-scale measures the free exchange of ideas and feelings between parent and children. The problems in family communication sub-scale measures the willingness of parents and children to honestly express their true thoughts and feelings to each other. Each item is scored on a 5-point Likert scale from 1 (strongly disagree) to 5 (strongly agree) in the original version. In order to avoid neutral attitude feedback, we adjusted all items to a 4-point Likert scale (1 = strongly disagree, 2 = disagree, 3 = agree, 4 = strongly agree) in the current study. The total PACS score ranges from 20 to 80, and higher scores mean better communication. Via the study of 3349 adolescents (age range of 12–15 years) from 35 secondary schools in China, the Cronbach alpha of this Chinese version scale is 0.88, and the results of confirmatory factor analysis found an acceptable fit of a model (**χ**^2^ = 2597.81, *df* = 146, **χ**^2^/df = 17.79, RMSEA = 0.07, CFI = 0.88) [30,31].

### 2.3. Measurement of Children’s Psychological Resilience

The psychological resilience of children was measured using the Chinese version of the Connor-Davidson Resilience Scale (CD-RISC) [32]. The English version consists of 25 items which assess psychological well-being in five dimensions: tenacity, tolerance of negative affect, positive acceptance of change, control, and spiritual influences. The Chinese version kept these 25 items, and revised items in three dimensions, namely tenacity, strength, and optimism. Each item is scored on a 5-point Likert scale from 1 (never) to 5 (often). Each dimension was measured by the summed score of its items as a sub-scale. A total score was calculated by summing the scores of three dimensions (thus ranging from 25 to 125). Higher scores indicate better psychological resilience. In a study carried out in the south province of China, the Cronbach alpha of this Chinese version scale (0.928), and test–retest reliability (r = 0.812, *p* < 0.001) and split-half reliability (r = 0.890, *p* < 0.001) were good [33].

### 2.4. Measurement of Psychological and Behavioral Status of Children

The psychological and behavioral status of students were evaluated by the Chinese version of Strength and Difficulties Questionnaires (SDQ) [34,35]. The scale contains five dimensions: emotional symptoms, conduct problems, hyperactivity, peer problems, and pro-social behavior. Each dimension has five items, and each item is scored on a 3-point Likert scale from 0 (totally noncompliant) to 2 (fully compliant). The total difficulties score is the sum of all dimension scores except the pro-social dimension, and a higher score reflects increased severity of emotional and behavioral problems. The reliability test from a study in the southeast province in China, showed that the Cronbach coefficient was 0.79, and the validity test showed the differences of the factors and total scores between experimental groups and control groups were significant (t = 2.07~6.31, *p* < 0.05) [35].

### 2.5. Covariates

Covariates in this study included gender, age, grade, sibling and self-reported family economic status relative to others in their community (much better off/better off/the same, poorer/much poorer). Most current studies divide the dissemination of their survey into two categories: left-behind children and never left-behind children. Since the majority of migration flows are within the country, it is not uncommon that migrant parents return home after living apart from their child for an extended period. The effects on children after the return of parents are also worth noting [36]. In order to have a more sensitive comparison of the status of parental migration, children in our study were divided into three groups: G1 (currently left-behind children); G2 (previously left-behind children); G3 (never left-behind children). They were asked to answer two questions: “has your father (or mother) taken a job away from your hometown and been absent for over six months?” The options were “yes, currently absent,” “yes, previously absent,” and “no, never.” If one or both parents were currently absent, the student was defined as G1; if one or both parents were previously absent, the student was defined as G2; and if neither parent was ever away, the student was G3.

### 2.6. Statistical Analysis

Data were analyzed using SPSS 23.0 (SPSS Inc., Chicago, IL, USA). First, sample characteristics were compared by chi-square test (for categorical variables) or analyses of variance (for continuous variables) among the three groups of children with different parental migration statuses. Second, structural equation models were used to evaluate the mediation hypothesis, and test that parental migration status and parent communication mediate psychological and related behavioral status through psychological resilience. Because these scores were not normally distributed, the differences in PACS and CD-RISC scores among the three parental migration status groups were compared by Kruskal-Wallis test [37]. Third, we set up three mediated models divided by parental migration status (G1, G2 and G3) in order to explore the effect of parental migration status and parent–child communication on mental health through psychological resilience. These models were tested with SPSS AMOS22.0 software using the Maximum Likelihood (ML) iteration procedure. The fit indices for a good model included: (1) ratios of **χ**^2^ value to the degrees of freedom of between 2 and 5; (2) comparative fit index (CFI) and Tucker-Lewis Index (TLI) > 0.95; and (3) root mean square error of approximation (RMSEA) < 0.05. Data were examined for normal distribution and statistical tests were two-tailed.

## 3. Results

Of the 1992 participants, 1251 were in G1, 473 were in G2, and 268 were in G3. Demographic characteristics are shown in Table 1. More than half of the subjects were male. The ages of the three groups are similar. More than half of the subjects were in grades 7 or 8. More than 60% reported their income level as “the same” as their community. Over 65% had siblings. About 70% of subjects described their parents’ education level as secondary school or below. (Table 1).

There were significant differences in the PACS, CD-RISC, and SDQ scores among the three groups of children with different parental migration status (Table 2). The total PACS scores were both highest in G3 (mother 56.02 ± 8.91; father 57.27 ± 10.11) and lowest in G2 (mother 54.07 ± 9.21; father 54.85 ± 10.52). The total CD-RISC score was highest in G3 (85.45 ± 17.14), and lowest in G1 (81.26 ± 15.83). The total SDQ score was highest in G1 (12.74 ± 5.37), and lowest in G3 (11.13 ± 5.09).

The SEM results showed that the fit for each of the three models was acceptable. All of CFI and TLI > 0.93, and RMSEA < 0.074 (see Table 3). Figure 1, Figure 2 and Figure 3 show the direct effects of parental migration status and parent–child communication on psychological resilience. In G1, currently left-behind status exhibited a negative effect (SSCs = −0.08, *p* < 0.001) on CD-RISC, and PACS scores (father SSCs = 0.21, *p* < 0.001; mother SSCs = 0.23, *p* < 0.001) were positively associated with CD-RISC scores. Currently left-behind status had no significant correlation to PACS scores (*p* > 0.05). In G2, PACS scores (father SSCs = 0.21, *p* < 0.001; mother SSCs = 0.23, *p* < 0.001) were positively associated with CD-RISC scores. Previously left-behind status has a negative effect (SSCs = −0.06, *p* < 0.001) on PACS scores, and no significant correlation to CD-RISC scores (*p* > 0.05). In G3, never left-behind status plays a positive role (SSCs = 0.06, *p* < 0.01) on CD-RISC scores, and PACS scores (father SSCs = 0.20, *p* < 0.001; mother SSCs = 0.23, *p* < 0.001) were positively associated with CD-RISC. Never left-behind status had no significant correlation to PACS (*p* > 0.05). In all three models, the direct effect of CD-RISC on SDQ was significant (SSCs = −0.21, *p* < 0.001).

## 4. Discussion

This study measured the score of PACS, CD-RISC, and SDQ among three groups of children with varying statuses of parental migration. Compared with the never left-behind group, children who were currently or previously left behind had relatively lower PACS and CD-RISC scores, and higher SDQ scores. Similar findings have been reported in previous studies, e.g., Su et al. found that children with one or two migrating parents reported the lowest levels of satisfaction and psychological adjustment [25]. It is worth noting that children whose parents were previously absent demonstrated the lowest PACS scores; this is the first study to report such a finding. It is possible that the experience of parental absence at a young age affects the formation of effective parent–child communication processes. This study also demonstrated that children with currently migrating parents have the lowest CD-RISC and the highest SDQ scores. This is consistent with other research findings, namely that left-behind children were significantly more likely to display externalizing and internalizing problems [38,39].

The structural equation model shows how parental migration status and parent–child communication affect the children’s psychological and related behavioral characteristics through psychological resilience. Our findings suggest that psychological resilience is a mediating factor among parental migration status, parent communication and psychological problems. The absence of parents has a negative effect on psychological resilience, while good parent communication has a positive effect. Our data also suggests that better psychological resilience can reduce psychological problems among different parental migration statuses. It seems that resilience may play a key role in helping left-behind children to maintain psychological wellbeing. There are few studies which use structural equation models to analyze the mediating role of psychological resilience, and this is the first study to report such a finding by comparison of the three parental migration status models. Ye et al., reported that resilience was found to be a protective factor for depressive symptoms and also mitigated the effects of peer victimization on depressive symptoms among rural-to-urban migrant children in China [40]. Therefore, resilience-based interventions might be useful to enhance the mental health of left-behind children, especially for currently left-behind children.

Our results showed that currently left-behind status had a negative influence on psychological resilience, while never left-behind status had a positive effect on psychological resilience. These results suggest that separation from parents does play a role in children’s mental health. This is echoed in other research on the subject: children who were separated from parents at a younger age had more symptoms of anxiety and depression [41,42]. While most studies in this area divided participants into left-behind children and never left-behind children, this study divided participants into currently left behind, previously left-behind and never left-behind groups. This allows for greater specificity in different statuses of parental migration.

This study demonstrated that parent–child communication has almost the same effect on psychological resilience under different parental migration status, and this conforms with previous research on this subject. Communication between parents and children is usually considered as an important factor, and similar findings have also been reported in other studies. Van et al., (2015) reported that parent–child communication was a promising factor to focus on in interventions aimed at preventing mental illness, and Elgar et al., (2013) reported that parent–child communication during family dinners had 13–30% positive effect on mental health [43,44]. We also found that communication with mothers was slightly more correlated with psychological resilience than was communication with fathers. The reason may be that mothers play a crucial role in fostering children, and most children prefer to communicate with their mothers than with their fathers [45]. As prior studies have suggested, good and regular communication with mothers is important in maintaining secure attachment between children and their absent mothers [21]. Children who have secure attachment relationships with their mothers may exhibit more psychological resilience.

Contrary to prior studies, the current study shows that there is no consistent correlation between parental migration status and parent–child communication. We found only that being previously left-behind had a slightly negative correlation to parent–child communication. Most prior studies indicated that the absence of parents does effect communication duration and frequency, creating a lack of social support for their children and leading to psychological problems [1,20,36]. We speculate that children who experience being left behind in their early childhood experience changes in their ability to communicate with their parents. Currently left-behind children are relatively older, and have regular communication with their parents. Due to better communication conditions (such as phone calls, video chats, etc.) in China, communication between migrant parents and left-behind children is easier now compared to the past [26]. Therefore, lack of timely communication might be a less significant influence than previously reported, and this topic warrants further research. However, this study did not measure the exact time and length of parental absence among children. Therefore, it is impossible to further judge the effect of separation length on parent–child communication. One study supports our findings to some extent: Hedenbro and Rydelius (2019) indicate that early child–mother–father communication was related to children’s social competence at the age of 15 [46].

The limitations of this study include, firstly, that we did not measure the time and duration of parental absence, so we cannot compare the differences among children who were left behind at different stages of childhood. Secondly, the Parent-Adolescent Communication Scale measures subjective feelings, and children who are left behind for a long time will have a bias in this area. They may mistakenly report their present status of communication with their parents as normal and good, as it may be better than it was when their parent was absent, even if it does not live up to more common standards of parental communication among those whose parents have never migrated or migrated for shorter periods of time. Thirdly, our study considered only a limited range of potential determinants: we did not explore areas such as children’s relationship with their caretakers and other related factors (i.e., family social capital, family context, etc.) Fourthly, our results do not make a comparison with left behind children whose parents migrate to another country.

## 5. Conclusions

This manuscript highlights that psychological resilience is the key mediating factor associated with parental migration status and parent–child communication. Better psychological resilience is related to fewer psychological problems among different parental migration statuses. To promote the health of left-behind children, interventions are needed to enhance psychological resilience, such as implementing positive psychology education in school, fostering their communication skill with parents, enhancing cultural adaptation self-efficacy [47], which may prevent psychological and related behavioral problems.

## Figures and Tables

**Figure 1 ijerph-18-05123-f001:**
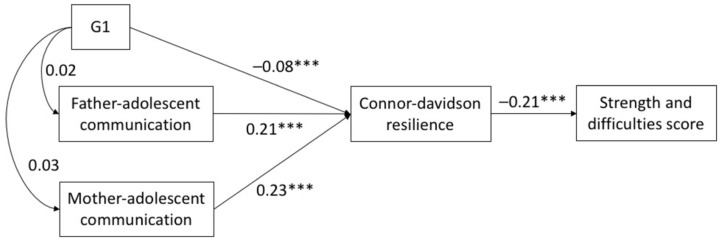
Structural equation model with the standardized coefficients among G1, parent-child communication and psychological resilience. *** *p* < 0.001.

**Figure 2 ijerph-18-05123-f002:**
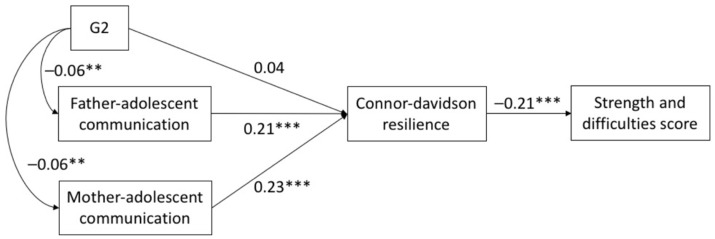
Structural equation model with the standardized coefficients among G2, parent-child communication and psychological resilience. ** *p* < 0.01, *** *p* < 0.001.

**Figure 3 ijerph-18-05123-f003:**
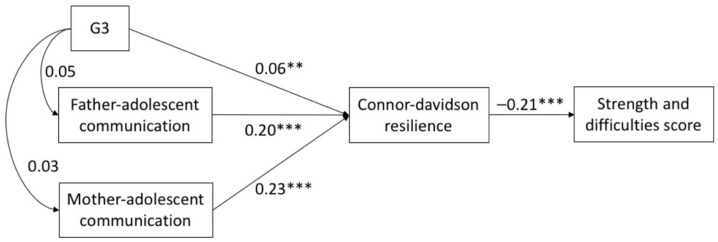
Structural equation model with the standardized coefficients among G3, parent-child communication and psychological resilience. ** *p* < 0.01, *** *p* < 0.001.

**Table 1 ijerph-18-05123-t001:** Sample characteristics n (%).

	G1	G2	G3	F or χ^2^	*p* Value
Gender				1.08	0.584
Male	678 (54.9)	257 (55.0)	137 (51.5)		
Female	558 (45.2)	210 (45.0)	129 (48.5)		
Age, mean(SD)	13.1 (1.2)	13.2 (1.2)	13.0 (1.2)	4.25	0.014
Grade				7.76	0.021
Grade 5 to 6	538 (43.0)	184 (38.9)	132 (49.4)		
Grade 7 to 8	712 (57.0)	289 (61.1)	135 (50.6)		
Perceived income level compared to community				5.94	0.204
Much better off /better off	317 (25.6)	120 (25.7)	85 (32.0)		
The same	818 (65.9)	306 (65.5)	165 (62.0)		
Poorer/much poorer	106 (8.5)	41 (8.8)	16(6.0)		
Any siblings				3.69	0.158
Yes	820 (65.6)	314 (66.4)	192 (71.6)		
No	431 (34.5)	159 (33.6)	76 (28.4)		
Mother education level				3.11	0.045
Primary school and below	424 (33.9)	184 (38.9)	93 (34.7)		
Secondary school	548 (43.8)	191 (40.4)	98 (36.6)		
Senior high school and above	104 (8.3)	44 (9.3)	39 (14.6)		
Unknown	175 (14.0)	54 (11.4)	38 (14.2)		
Father education level				5.23	0.005
Primary school and below	288 (23.0)	130 (27.5)	56 (20.9)		
Secondary school	677 (54.1)	226 (47.8)	115 (42.9)		
Senior high school and above	134 (10.7)	65 (13.8)	61 (22.8)		
Unknown	152 (12.2)	52 (11.0)	36 (13.4)		

Note: G1 (currently left-behind children); G2 (previously left-behind children); G3 (never left-behind children).

**Table 2 ijerph-18-05123-t002:** Comparisons among parental migration status regarding parent–child communication, psychological resilience score, and strength and difficulties questionnaires score, mean (SD).

	G1	G2	G3	H
Mother-adolescent communication score	55.4 (9.7)	54.1 (9.2)	56.0 (8.9)	11.82 **
Father-adolescent communication score	56.1 (10.2)	54.9 (10.5)	57.3 (10.1)	8.41 *
Psychological resilience score	81.3 (15.8)	82.4 (15.3)	85.5 (17.1)	14.93 ***
Total difficulties score	12.7 (5.4)	12.5 (5.2)	11.1 (5.1)	21.14 ***

* *p* < 0.05, ** *p* < 0.01, *** *p* < 0.001.

**Table 3 ijerph-18-05123-t003:** The fit indices of the multigroup models.

Models	χ^2^	df	χ^2^/df	CFI	TLI	RMSEA
G1	170.156 ***	10	17.02	0.994	0.941	0.071
G2	168.842 ***	10	16.88	1.000	1.000	0.004
G3	170.020 ***	10	17.00	0.994	0.936	0.074

*** *p* < 0.001.

## Data Availability

Data can be requested from the corresponding author.

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
