# Peer review of "Left-Behind Children, Parent-Child Communication and Psychological Resilience: A Structural Equation Modeling Analysis"

_ijerph, 2021, doi:10.3390/ijerph18105123_

Round 1
Reviewer 1 Report
Thank you for this interesting study. The experience of left-behind children warrants exploration due to its peculiar nature and adverse effects on child’s development.
To my opinion, this manuscript is generally well written although some English language editing is recommended.
Following are my comments - more content-related:
- Abstract: p. 1, lines 16/7: the names of the instruments should be followed by acronyms into brackets. Then, acronyms can stand alone.
- Introduction:
Please highlight the peculiar nature of the left-behind experience, and differences with children who are left-behind (highly neglected) for reasons other than parental migration due to work. If it is true that both experiences can have adverse effects, they cannot be considered similar, and the traumatic nature itself is highly different.
Can you provide a definition of resilience drawn by existing literature?
- 2, lines 70/1: ‘Therefore, the present parent-child communication may more influenced by parents’ will, but not the poor communication condition’. This concept should be fully explained and English revised.
- Method:
- 3, lines 112/3: ‘The total PACS score ranges from 10 to 40’. I am not sure. Instead, did you refer to each subscale score range? Moreover, I would specify the number of items for each subscale.
- 3, lines 140/1: I wonder whether ‘abnormal’ and ‘normal’ samples may be indicated as ‘experimental’ and ‘control’ groups.
- 4, lines 146/7: ‘In order to have a more sensitive comparison of the status of parental migration, children in our study were divided into three groups’. This aspect should be explained. What do you mean exactly with ‘sensitive comparison’? I can understand it, but I think that readers can benefit from a fully explained hypothesis for introducing a third group.
- 4, lines 149-151: ‘They were asked to answer two questions: has your father (or mother) taken a job away from your hometown and been absent for over six months?’ What was the reason for choosing a time period of six months? Was it literature-driven?
- 4, lines 163/4: can you provide the reference of the Kruskal-Wallis test?
- Results: p. 4, line 173: G1, G2, G3 sample sizes are not well balanced. Please, mention this element among study limitations.
- Discussion: p. 7 lines 257-259: ‘We also found that communication with mothers was slightly more correlated with psychological resilience than was communication with fathers’. I wonder whether the authors can speculate more about this difference although minimal. At the end of the paragraph about study limitations, I would suggest to highlight the narrow scope of your study, where the family context was barely taken into account.
- Conclusion: I would suggest to mention few examples of interventions focused on enhancing resilience during childhood and adolescence in this context, if already available in literature.
A minor suggestion:
- Title: please consider if ‘Left-behind experience’ may be substituted by ‘Left-behind children’.
Reviewer 2 Report
This article addresses a topic of undoubted interest, that of the impact of migratory processes on children's development and mental health. Although this is not about international migrations, this study can contribute knowledge to the emerging lines of research in transnational families, in which the children are educated by caregivers as a result of the migration of the parents, with whom they maintain relationships basically by way. telematics. And judging by the differences in the number of subjects between G1 and G2 on the one hand and G3 on the other, it can be interpreted that the migration of parents leaving their children in the studied region is in fact the norm, with which their study it is relevant to the intervention.
The state of the art reviewed by the authors is complete and relevant to the research problem posed. The methodology is adequate, it is applied correctly and it is perfectly explained. The results are not surprising, and they corroborate what is stated in the state of the art. The contribution of the measure of resilience as a mediating factor between immigration status and behavior problems is remarkable.
As a weak point is the undifferentiation in the migratory status of parents between those with both absent parents, with only the absent father or with only the absent mother. In the article, it does not refer at any time to the forms of emigration in the local context that it studies, whether they are fundamentally male, female or the migration of the complete couple predominates. I consider it relevant to report on this detail. In the first place, because it is not the same that both parents are absent, that only one is absent. Second, we do not know if there are gender differences, if the absence of the father or the mother is significantly different in relation to the areas studied. And, thirdly, making this dimension explicit would have been useful for comparison with international studies. Thus, in Latin America, there is a predominant migration profile whereby it is the mother who emigrates leaving her children in the care of relatives, being relevant whether it is a single parent family or not. Comparing the results with the processes in different countries would have been a relevant contribution. Therefore I would suggest adding this to the limitations of the study.
Otherwise it can be published in its current form, only stating the indicated limitation.
